# OMNICONTRAST: VISION-LANGUAGE-INTERLEAVED CONTRAST FROM PIXELS ALL AT ONCE

## ABSTRACT

In this work, we present OmniContrast, a unified contrastive learning model tailored for vision, language, and vision-language-interleaved understanding within multi-modal web documents. Unlike traditional image-caption data with clear vision-language correspondence, we explore a new contrastive fashion on maximizing the similarity between consecutive snippets sampled from image-text interleaved web documents. Moreover, to enable CLIP to handle long-form text and image-text interleaved content from web documents, OmniContrast unifies all modalities into pixel space, where text is rendered visually. This unification simplifies the processing and representation of diverse multi-modal inputs, enabling a single vision model to process any modality. To evaluate the omni-modality understanding of OmniContrast, we design three consecutive information retrieval benchmarks AnyCIR, SeqCIR, and CSR. Extensive experimental results demonstrate that OmniContrast achieves superior or competitive omni-modality understanding performance to existing standard CLIP models trained on image-text pairs. This highlights the potential of multi-modal web documents as a rich and valuable resource for advancing vision-language learning.

## 1 INTRODUCTION

Learning vision-language correspondence from image-caption pairs, particularly with the advent of contrastive learning methods like CLIP (Radford et al., 2021), has made significant strides in multi-modal research. These models exhibit strong zero-shot cross-modal ability across various downstream tasks (Gu et al., 2021; Ramesh et al., 2021; Wortsman et al., 2022) due to their vision-language aligned representation space.

However, most CLIP-style models face challenges in understanding complex multi-modal information correspondence under web document retrieval scenarios. As shown in Fig. 1, web doc-

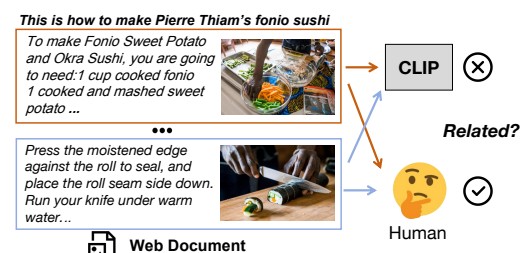

**Figure 1:** Modeling implicit vision-language correspondence within the same multi-modal document is challenging for existing CLIP models as they are solely trained on image and directly aligned captions.

uments often consist of loosely related image-text interleaved content and long-form text, while CLIP models are primarily trained on images and directly aligned short captions. Although efforts have been made to develop universal multi-modal embedding with various text (Wei et al., 2023; Jang et al., 2024) or to handle long-form caption input (Zhang et al., 2024; Zheng et al., 2024) for CLIP models, *direct training of CLIP on multi-modal interleaved documents for omni-modality representation remains uncharted.* To design such a new contrastive learning paradigm, it is essential to first define what constitutes contrast within image-text interleaved documents and how to effectively represent the omni-modal input, especially for long text and being interleaved.

To address these challenges, we present OmniContrast, which unifies the image, text, and image-text interleaved modalities from multi-modal web documents in contrastive learning by representing all inputs in pixel space, as shown in Fig. 2. For contrast target, OmniContrast aligns two consecutive multi-modal snippets from the same document by maximizing their embedding similarity. Each snippet can consist of image-only, text-only, or image-text interleaved content. The consecutive doc-

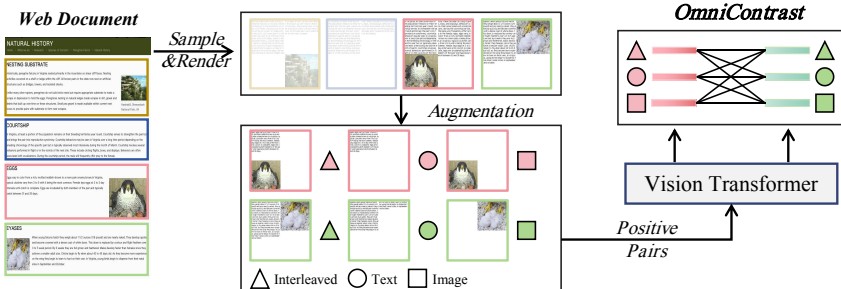

**Figure 2:** OmniContrast explore an alternative vision-centric paradigm for unifying vision-language modeling from image-text interleaved web data. It uses a single vision transformer to process any modality presented in pixels and thereby natively learn a unified representation for omni-modalities.

ument snippets exhibit a loose yet reasonable vision-language correspondence. Generally, images often convey critical information that enhances the readability and understanding of coherent text paragraphs in multi-modal web documents. Moreover, we design the modality masking and text masking data augmentation strategy to improve the diversity of training data.

To seek a unification of omni-modality representation, OmniContrast unify all input into pixel space by rendering text into images. Specifically, we represent all modality data as a 2×2 grid image, where each grid can be visual text or image content. Since image-text interleaved content is primarily presented in visual form on the web, pixel space provides a natural fit for representing image-text interleaved data. Additionally, as shown by CLIPPO (Tschannen et al., 2023), the visual text can convey longer context while keeping linguistic semantics in contrastive learning. Consequently, unifying all data in pixel space simplifies pre-processing and reduces the need for specialized model designs to handle omni-modal data. We provide a more detailed discussion in Sec. 6.

Moreover, we design AnyCIR benchmark to evaluate the cross-modality information retrieval under the omni-modalities context and SeqCIR benchmark to assess the fine-grained consecutive relationship modeling within documents by retrieving consecutive snippets sequentially. To evaluate the transferability of OmniContrast in real-world scenarios, we further design a zero-shot consecutive slide retrieval (CSR) benchmark, where slides are more complex image-text interleaved data. Our extensive experiments also show that OmniContrast can achieve superior zero-shot multi-modal information retrieval on M-BEIR (Wei et al., 2023) and text embedding learning on MTEB (Muennighoff et al., 2023). Additionally, we also investigate the impact of various contrast targets (image-caption, consecutive and non-consecutive snippets) and observe that joint image-text interleaved training can further improve language understanding in pixel space.

**Contributions.** our contributions are three-folds: 1). To the best of our knowledge, OmniContrast is the first to explore vision-language correspondence on image-text interleaved web documents in CLIP-style. 2). OmniContrast is a single unified vision model with advanced vision, language, and vision-language interleaved modality understanding capacity from pixel space for multi-modal web document retrieval scenarios. 3). To facilitate the evaluation of omni-modality understanding, we propose three consecutive information retrieval benchmarks, including AnyCIR, SeqCIR, and CSR. Moreover, our extensive experimental results show that OmniContrast achieve superior performance in our proposed consecutive information retrieval benchmarks, zero-shot multi-modal information retrieval benchmark M-BEIR, and text embedding learning benchmark MTEB.

## 2 RELATED WORK

### 2.1 VISION-LANGUAGE LEARNING FROM WEB DATA

The pioneer work CLIP (Radford et al., 2021) establishes a breakthrough learning paradigm by applying contrastive learning on large-scale noisy image/alt-text paired data from the internet. Follow-up studies scale the image-text pairs data (Schuhmann et al., 2022; Gadre et al., 2024) and the model design (Li et al., 2022; Yu et al., 2022; Zhai et al., 2023) to further improve the performance. More recently, with the rapid development of Multi-modal Large Language Models (MLLMs) (Li et al., 2023; Liu et al., 2024; Lin et al., 2024), multi-modal web documents data, such as MMC4 (Zhu et al., 2024) and OBELICS (Laurençon et al., 2024), have emerged as new sources of training data. These

multi-modal documents typically consist of sequences of coherent text paragraphs interleaved with images. Several research (Lin et al., 2024; McKinzie et al., 2024) demonstrate that joint training with image-text data and multi-modal web documents outperforms solely image-text pairs, which indicates the multi-modal documents contain useful vision-language correspondence from image-text pairs. Moreover, (Ma et al., 2024; Lu et al., 2024; Jang et al., 2024) leverage MLLMs to encode multi-modal document information for question answering or document retrieval. In contrast to prior research focusing on MLLMs, we serve as the first step in studying the potential of contrastive learning on image-text interleaved web document data.

## 2.2 VISUAL REPRESENTATION FOR LANGUAGE MODELING

Despite the impressive results achieved by text tokenization (Devlin, 2018; Sennrich, 2015) in language modeling (Devlin, 2018; Brown, 2020), text tokenization is vulnerable to text permutations (Salesky et al., 2021), such as misspellings and has limited scalability to other languages (Rust et al., 2022). To address these challenges, a line of works explores the tokenizer-free solution based on the visual representation of text. (Meng et al., 2019) use glyph-vectors from Chinese characters images to enhance the text representation. (Salesky et al., 2021) proposed visual text representation as open-vocabularies to improve the robustness of machine translation. Recently, to close the gaps between the visual text representation and text tokenization, (Rust et al., 2022; Xiao et al., 2024; Gao et al., 2024; Chai et al., 2024) further explore different pre-training strategies, such as next patch prediction, next token prediction, and contrastive learning.

In the vision-language domain, the most closely related work is CLIPPO (Tschannen et al., 2023). CLIPPO utilizes rendered alt-text and image pairs to train the vision encoder using contrastive learning the same as CLIP. *In contrast, OmniContrast marks the first attempt at exploration in image-text interleaved documents contrastive learning and omni-modality learning.* Additionally, screenshot understanding (Gao et al., 2024) is also closely related to visual text representation learning, which involves language modeling from documents (Kim et al., 2022), web pages (Lee et al., 2023) or UI images (Li & Li, 2022). Despite these screenshot language models directly learning text information from the input image, they still can not handle omni-modality input.

## 3 OMNICONTRAST

As shown in Fig. 2, OmniContrast uses rendered consecutive snippets sampled from multi-modal web documents as training data. After data pre-processing and augmentation, each snippet in positive pairs can be either image-only, text-only or an interleaved image-text rendered image. During training, the single vision model is optimized by contrastive loss on these consecutive data pairs.

### 3.1 INTERLEAVED WEB DATA PROCESSING

**Document Pre-processing.** Given a web document, our goal is to sample a pair of semantically relevant image-text snippets for training. Firstly, we split a document text into multiple text segments with a maximum of 1,100 characters in each segment. Then, we use the CLIP similarity annotation provided in MMC4 dataset (Zhu et al., 2024) to assign the image to the corresponding segments. Each interleaved snippet at least contains text while can be without images or assigned multiple images. For the multiple image cases, we only randomly sample one image for training.

**Data Augmentation.** Next, we apply two types of augmentations to obtain augmented snippets, i.e., *modality masking* and *text masking*. In modality masking, we only mask snippets with both text and image contents. During training, we apply modality masking with a masking rate of 40% on snippets to randomly drop one modality content. With modality masking, we are able to sample diverse training matching targets. For text masking, we randomly remove sentences from the beginning or end of the text content in 40% of the snippets. This augmentation enhances the model's language understanding by preventing the model from overfitting on recurring words.

**Multi-modal Snippet Rendering.** Given a multimodal snippet containing both image and text, we render its content into a 2×2 grid. Each grid has a resolution of 224×224 pixels. If the snippet includes an image, we resize it to fit the grid and place it in a randomly selected grid cell. For visual text rendering, we follow the approach in (Tschannen et al., 2023) using the GNU Unifont

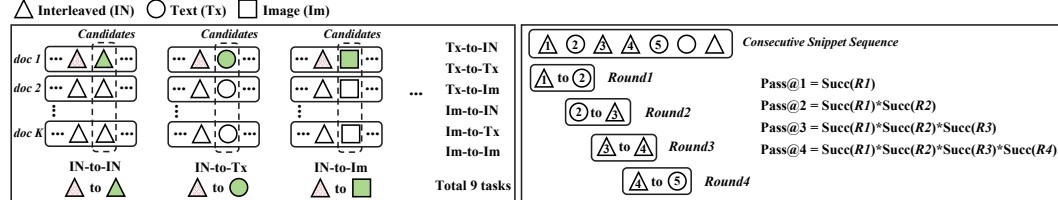

**(a). AnyCIR Benchmark**        **(b). SeqCIR Benchmark**

**Figure 3: (a):** In AnyCIR, we first sample consecutive snippet pairs from distinct documents and use the former snippet to retrieve the latter one. For each query, we use all the later snippets as candidates. The combination of different modalities results in 9 retrieval tasks in total. **(b):** In SeqCIR, we sequentially retrieve the consecutive snippets in multiple rounds. For each query, we use all the snippets segmented from 5k documents as candidates. For each query, we ignore the preceding snippets in the previous round.

bitmap font. The long-form text can be rendered across multiple grids, starting from the top-left and proceeding left-to-right and top-to-bottom. Once one grid is fulfilled with either image or text content, the rendering process continues in the next available grid.

### 3.2 TRAINING OBJECTIVES

**Positive Pairs Sampling.** After data pre-processing, a document $d_i$ is segmented as a serials of snippets, i.e., $\{s_i^n\}_{n=0}^N \in d_i$. During training, we sample snippet pairs $(s_i^q, s_i^k)$ from the same documents $d_i$ as positive pairs, while the snippets from other documents are negative terms. We use consecutive snippets, i.e., $k = q + 1$, to construct positive pairs as our default setting. To ablate the optimal training targets, we also investigate the sampling strategy of pairs with one-hop distance, i.e., $k = q + 2$. To differentiate, we use **Omni** to denote consecutive pairs only, and **Omni+/++** to denote 20%/40% of pairs are sampled from one-hop distance pairs.

**Contrastive Learning.** Our training objective is contrastive loss (Oord et al., 2018) formulated as,

$$\mathcal{L}_c = -\frac{1}{N} \sum_{i=1}^N log \frac{exp(f_i^q \cdot f_i^k)/\tau)}{\sum_{j=1}^N exp(f_i^q \cdot f_j^k)/\tau)}, \tag{1}$$

where $(f_i^q, f_i^k)$ is the visual features extracted from sampled snippets $(s_i^q, s_i^k)$ from the same document $d_i$ and $\tau$ is the temperature to control the sharpness of the logit distribution.

## 4 CONSECUTIVE INFORMATION RETRIEVAL

To evaluate the consecutive information retrieval capabilities, we design two multi-modal snippet retrieval benchmarks based on OBELICS (Laurençon et al., 2024) and zero-shot slide retrieval based on Slideshare-1M (Araujo et al., 2016). Compared to the training dataset MMC4, the OBELICS preserves the original image text interleaved order, which is closer to real-world scenes. The slides in Slidershare-1M are naively interleaved multi-modal data with more complex interleaved forms.

**Any-to-Any Consecutive Information Retrieval (AnyCIR).** In this task, we aim to retrieve any modality consecutive information given any modality queries, as shown in Fig. 3(a). The types of modality include interleaved (**IN**), Text only (**Tx**), and Image only (**Im**), resulting in 9 tasks in total with different combinations. The AnyCIR consists of 20,000 randomly sampled consecutive snippet pairs from distinct documents. Each snippet in the pair includes text and at least one image content. During inference, all the tasks share the same snippet pair source. For retrieval tasks with a single modality, we simply mask other modalities during rendering. We render images into a randomly chosen grid for both queries and candidates.

**Sequential Consecutive Information Retrieval (SeqCIR).** This task aims to evaluate the fine-grained consecutive information modeling capacity. For each query, the candidate pool consists of 26,433 snippets from 5,000 distinct documents. For each snippet, we use the full text and one randomly selected image if applicable. We use 2,524 snippets as the initial query set, which are the first snippets of the documents. For this task, we iteratively retrieve the next consecutive snippets and only successful retrieval queries are passed to the next iteration. For each iteration, we ignore the preceding snippets of the query snippet in the documents. The Pass@k rate denotes the success rate of sequential retrieval at the $n^{th}$ round, as shown in Fig. 3(b). The SeqCIR is a very challenging

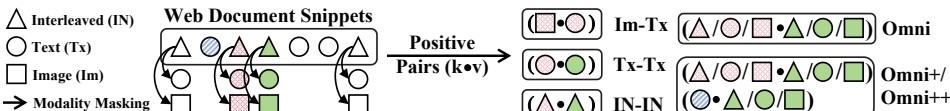

**Figure 4:** Illustration of positive contrastive pair settings of different baseline models.

task as the candidate pool of SeqCIR contains subsequent snippets from the same documents. It requires the model to accurately distinguish the most consecutive snippet.

**Zero-Shot Consecutive Slide Retrieval (CSR).** To better examine the transferability of Omni-Contrast under real-world scenario, we propose a benchmark of retrieving the most relevant slide. Specifically, we sample 28,016 pairs of consecutive slide images from Slideshare-1M (Araujo et al., 2016). Each pair is sampled from a distinct slide deck (more than 6 slides) after removing the first two slides. For evaluation, we use the former slide as a query and all the latter slides as candidates. Despite some consecutive slides might share similar layouts or part of content overlap, our experimental results show that it is still a challenging task even using these shortcuts.

# 5 EXPERIMENTS

## 5.1 EXPERIMENTAL SETUP.

**Data Variant Baselines.** To better understand the model capacity learned from interleaved data, we further construct different positive pair data as our baselines as illustrated in Fig. 4. Our baselines include **1).** Image-Text **(Im-Tx)** pairs sampled from a LAION subset; **2).** Image-Text **(Im-Tx)** pairs from the same snippet of MMC4, where we use the MMC4 annotation to generate the pairs, i.e. the CLIP similarity assignment; **3).** Text-Text **(Tx-Tx)** pairs by masking all the images in the snippets; **4).** Interleaved-Interleaved **(IN-IN)** pairs by sampling from the snippets pairs containing both image and text content; **5). Omni**$_{224}$ pairs first rendering in $448 \times 448$ resolution then resize to $224 \times 224$ resolution for fair comparison with original CLIP model; **6). Omni+/++** denotes 20%/40% of pairs are sampled from one-hop pairs. All baselines use the same training setting.

**Implementation Details.** Our implementation is based on OpenCLIP (Ilharco et al., 2021). In all experiments, we use ViT-B-16 (Dosovitskiy, 2020) with an input resolution $448 \times 448$. We use a batch size of 1024 and a learning rate of 1e-4 for training 20 epochs. Our pretraining dataset uses the MMC4-core-fewer-face (Zhu et al., 2024) subset, comprising 5 million documents with both images and text, totaling 17 million images. We use CLIP (Radford et al., 2021) checkpoint as our initialization due to the small scale of our training data.

## 5.2 CONSECUTIVE MULTI-MODAL INFORMATION RETRIEVAL

We include the vision encoder of CLIP (Radford et al., 2021), OpenCLIP (Cherti et al., 2023), and CLIPPO (Tschannen et al., 2023) in the model size of ViT-B as our baseline. Note that these baselines are trained on different sources and scales of image-text pair data.

**Any-to-Any Consecutive Information Retrieval (AnyCIR).** In Table 1, we report 9 retrieval task results at Rank@1 metric. It can be observed that image-text interleaved data can help the model better understand visual text data. For example, Omni and IN-IN models achieve better results on the Tx-to-Tx retrieval task than the Tx-Tx baseline. Moreover, more diverse training data can boost the performance of omni-modality representation learning, as Omni achieves better performance on the IN-to-IN task compared to the IN-IN baseline. When training the model with none-consecutive samples, i.e. Omni+ or Omni++, the performance only slightly decreases, which indicates that the close snippets generally have consistent vision-language correspondence. Additionally, Omin$_{224}$ indicates that our performance gains not only from the higher input resolution but also from our novel training data design. Interestingly, the CLIP vision encoder has stronger visual text understanding capacity over OpenCLIP which is trained on a larger scale of datasets. When training on image-text pair data from LAION, the model performs poorly on the AnyCIR benchmark indicating the large domain gap between image-caption and multi-modal document data.

**Sequential Consecutive Information Retrieval (SeqCIR)**. Table 2 reports sequential consecutive snippets retrieval results in a total of four rounds. The best model only achieves a 3.7% success rate

**Table 1:** Any-to-Any Consecutive Information Retrieval benchmark on Rank@1 metric. The modalities include Image-Text Interleaved (**IN**), Text only (**Tx**), and Image only (**Im**). Gray results refer to the model input resolution as 224 and the default is 448.

| Model | Data | IN-IN | IN-Tx | IN-Im | Tx-IN | Tx-Tx | Tx-Im | Im-IN | Im-Tx | Im-Im | Overall |
|---|---|---|---|---|---|---|---|---|---|---|---|
| CLIP-V | WIT 400M | 24.10 | 6.18 | 5.27 | 14.23 | 11.47 | 1.02 | 11.60 | 0.93 | 12.45 | 9.69 |
| OpenCLIP-V | LAION 2B | 18.41 | 0.26 | 12.23 | 4.73 | 3.82 | 0.86 | 13.52 | 0.02 | 15.76 | 7.73 |
| CLIPPO | YFCC 100M | 10.17 | 0.01 | 9.99 | 0.00 | 0.01 | 0.01 | 6.31 | 0.02 | 11.79 | 4.25 |
| Omni$_{224}$ | MMC4-core | 69.39 | 67.20 | 13.89 | 67.86 | 70.61 | 5.04 | 14.00 | 5.68 | 14.45 | 36.45 |
| Im-Tx | LAION 40M | 25.64 | 15.23 | 11.89 | 21.21 | 26.40 | 5.72 | 15.07 | 5.36 | 16.20 | 15.86 |
| Im-Tx | MMC4-core | 63.34 | 59.15 | 15.60 | 61.30 | 61.08 | **12.34** | 17.36 | **12.31** | 17.97 | 35.60 |
| Tx-Tx | MMC4-core | 53.16 | 62.34 | 0.01 | 61.12 | 73.38 | 0.01 | 0.03 | 0.02 | 0.78 | 27.87 |
| IN-IN | MMC4-core | 76.56 | **74.85** | 0.40 | **74.19** | 74.81 | 0.12 | 2.58 | 0.64 | 8.95 | 34.79 |
| Omni | MMC4-core | **78.27** | 73.89 | **22.10** | 74.19 | 74.32 | 10.08 | **22.00** | 10.95 | 19.50 | **42.81** |
| Omni+ | MMC4-core | 77.94 | 73.68 | 21.87 | 73.73 | 73.68 | 10.06 | 21.76 | 10.70 | 19.29 | 42.52 |
| Omni++ | MMC4-core | 78.05 | 73.53 | 21.27 | 73.57 | 73.41 | 9.96 | 21.48 | 10.63 | **19.55** | 42.38 |

**Table 2:** Sequential Consecutive Information Retrieval. Pass@k denotes the retrieval success rate at $k^{th}$ round. Gray results refer to the model input resolution as 224 and the default is 448.

| Model | Data | Pass@1 | Pass@2 | Pass@3 | Pass@4 |
|---|---|---|---|---|---|
| CLIP-V | WIT 400M | 11.69 | 1.51 | 0.24 | 0.04 |
| OpenCLIP-V | LAION 2B | 7.49 | 0.71 | 0.16 | 0.00 |
| CLIPPO | YFCC 100M | 3.86 | 0.36 | 0.09 | 0.00 |
| Omni$_{224}$ | MMC4-core | 31.85 | 10.97 | 5.39 | 2.81 |
| Im-Tx | LAION 40M | 13.00 | 1.90 | 0.32 | 0.04 |
| Im-Tx | MMC4-core | 29.48 | 9.03 | 3.80 | 1.58 |
| Tx-Tx | MMC4-core | 26.39 | 7.21 | 3.01 | 1.55 |
| IN-IN | MMC4-core | 32.53 | 12.96 | 6.38 | 3.57 |
| Omni | MMC4-core | **34.43** | **13.07** | 6.78 | **3.76** |
| Omni+ | MMC4-core | 33.28 | 12.60 | 6.50 | 3.68 |
| Omni++ | MMC4-core | 33.76 | 12.56 | 6.42 | 3.76 |

**Table 3:** Zero-Shot Consecutive Slides Retrieval. Gray results refer to the model input resolution as 224 and the default is 448.

| Model | Data | R@1 | R@5 | R@10 | Avg |
|---|---|---|---|---|---|
| CLIP-V | WIT 400M | 34.60 | 45.10 | 49.29 | 43.00 |
| OpenCLIP-V | LAION 2B | **38.08** | **48.33** | **52.27** | **46.23** |
| CLIPPO | YFCC 100M | 26.42 | 34.31 | 37.30 | 32.68 |
| Omni$_{224}$ | MMC4-core | 33.81 | 43.28 | 47.02 | 41.37 |
| Im-Tx | LAION 40M | 26.21 | 33.13 | 35.85 | 31.73 |
| Im-Tx | MMC4-core | 34.68 | 43.45 | 46.85 | 41.66 |
| Tx-Tx | MMC4-core | 11.04 | 14.59 | 16.14 | 13.92 |
| IN-IN | MMC4-core | 25.92 | 33.40 | 36.46 | 31.93 |
| Omni | MMC4-core | 44.05 | **55.55** | **59.74** | 53.11 |
| Omni+ | MMC4-core | **44.21** | 55.54 | 59.68 | **53.14** |
| Omni++ | MMC4-core | 43.74 | 55.16 | 59.29 | 52.73 |

after four rounds, which indicates that these models still lack of capacity for fine-grained consecutive relation modeling. The results also draw the same observation as the AnyCIR benchmark, which is that diverse training data helps omni-modality representation learning.

**Zero-Shot Consecutive Slide Retrieval (CSR).** As shown in Table 3, the Omni model achieves the best results with 44% rank@1 accuracy under zero-shot setting. It indicates that our learned interleaved representation is able to generalize to the complex interleaved data, i.e. slide. Moreover, the results demonstrate that the language understanding capacity of OmniContrast can be generalized beyond rendered text to various styles and font sizes. We also find that OpenCLIP is better than CLIP in CSR, which is in contrast to previous benchmarks. One possible reason is that the OpenCLIP has been trained with slide data as suggested in (Lin et al., 2023).

## 5.3 Traditional Multi-modal Information Retrieval

To investigate the ability of OmniContrast in traditional information retrieval tasks, we adopt zero-shot M-BEIR (Wei et al., 2023) for evaluation, which assembles 10 diverse datasets from multiple domains with 8 distinct multi-modal retrieval tasks. In our setting, we render all modality information (image and text) into a single image for all the queries and candidates without using instructions. As we find out the balance of the modality information is critical to this task, we pad all the text input to 800 chars by repeating them. We provide the ablation study results on supply materials.

Table 4 shows the zero-shot union candidate pool results of OmniContrast and baselines, including CLIP$_B$(ViT-B), CLIP$_L$(ViT-L), SigLIP (Zhai et al., 2023), BLIP (Li et al., 2022) and BLIP2 (Li et al., 2023). OmniContrast using single vision encoder outperforms the models with separate text encoder under the zero-shot setting, e.g.SigLIP. Also, it can be seen that the models trained on interleaved data generally are good at WebQA (Chang et al., 2022) while performing poorly on InfoSeek (Chen et al., 2023) compared to the CLIP-style model. It indicates that the interleaved web data and image-caption data empower the model with different capacities.

## 5.4 Text Embedding Benchmark

To evaluate the language understanding capability, we use MTEB (Muennighoff et al., 2023) English subset which comprises 7 different tasks in a total of 56 datasets. During inference, we render

**Table 4:** Zero-shot results on M-BEIR$_{union}$ (Recall@5). Im-Tx$_{la}$ denote train on LAION 40M data.

| Task | Dataset | CLIP$_B$ | CLIP$_L$ | SigLIP | BLIP | BLIP2 | Im-Tx$_{la}$ | Im-Tx | Tx-Tx | IN-IN | Omni | Omni+ | Omni++ |
|---|---|---|---|---|---|---|---|---|---|---|---|---|---|
| 1. $q_t \rightarrow c_i$ | VisualNews | 0.0 | 0.0 | 0.0 | 0.0 | 0.0 | 0.2 | 0.1 | 0.0 | 0.0 | 0.2 | 0.2 | 0.2 |
|  | MSCOCO | 0.0 | 0.0 | 0.0 | 0.0 | 0.0 | 0.1 | 0.0 | 0.0 | 0.0 | 0.0 | 0.0 | 0.0 |
|  | Fashion200K | 0.0 | 0.0 | 0.0 | 0.0 | 0.0 | 0.1 | 0.0 | 0.0 | 0.0 | 0.0 | 0.0 | 0.0 |
| 2. $q_t \rightarrow c_t$ | WebQA | 32.5 | 32.1 | 34.0 | 38.1 | 35.2 | 35.9 | 47.3 | 41.0 | 46.0 | 46.2 | 48.5 | 49.3 |
| 3. $q_t$ $\rightarrow (c_i, c_t)$ | EDIS | 3.0 | 6.7 | 1.1 | 0.0 | 0.0 | 1.7 | 2.3 | 4.4 | 11.4 | 10.6 | 11.5 | 12.3 |
|  | WebQA | 0.8 | 5.5 | 2.1 | 0.0 | 0.0 | 1.2 | 6.8 | 24.0 | 40.7 | 27.4 | 29.1 | 29.5 |
| 4. $q_i \rightarrow c_t$ | VisualNews | 0.0 | 0.0 | 0.0 | 0.0 | 0.0 | 0.0 | 0.2 | 0.0 | 0.0 | 0.2 | 0.3 | 0.2 |
|  | MSCOCO | 0.0 | 0.0 | 0.0 | 0.0 | 0.0 | 0.1 | 0.2 | 0.0 | 0.0 | 0.3 | 0.3 | 0.3 |
|  | Fashion200K | 0.0 | 0.0 | 0.0 | 0.0 | 0.0 | 0.0 | 0.0 | 0.0 | 0.0 | 0.0 | 0.0 | 0.0 |
| 5. $q_i \rightarrow c_t$ | NIGHTS | 27.1 | 25.3 | 28.7 | 25.1 | 24.0 | 28.0 | 27.1 | 0.2 | 15.7 | 25.0 | 24.3 | 25.5 |
| 6. $(q_i, q_t)$ $\rightarrow c_t$ | OVEN | 0.0 | 0.0 | 0.0 | 0.0 | 0.0 | 0.0 | 0.3 | 0.0 | 0.1 | 0.6 | 0.6 | 1.0 |
|  | InfoSeek | 0.0 | 0.0 | 0.0 | 0.0 | 0.0 | 0.0 | 0.3 | 0.0 | 0.0 | 0.2 | 0.2 | 0.4 |
| 7. $(q_i, q_t)$ $\rightarrow c_i$ | FashionIQ | 1.0 | 4.4 | 4.8 | 2.2 | 3.9 | 6.8 | 2.7 | 0.0 | 0.5 | 3.8 | 4.2 | 3.5 |
|  | CIRR | 1.6 | 5.4 | 7.1 | 7.4 | 6.2 | 7.4 | 3.1 | 0.0 | 0.2 | 5.5 | 5.9 | 5.7 |
| 8. $(q_i, q_t)$ $\rightarrow (c_i, c_t)$ | OVEN | 1.0 | 24.5 | 27.2 | 10.1 | 13.8 | 14.5 | 2.2 | 0.0 | 0.1 | 5.8 | 6.1 | 4.8 |
|  | InfoSeek | 0.6 | 22.1 | 24.3 | 7.9 | 11.4 | 11.1 | 1.7 | 0.0 | 0.2 | 4.2 | 4.6 | 3.1 |
| - | Average | 4.2 | 7.9 | 8.1 | 5.7 | 5.9 | 6.7 | 5.9 | 4.3 | 7.2 | 8.1 | **8.5** | **8.5** |

**Table 5:** Mass Text Embedding Benchmark. The rows in Cyan refer to the text encoder directly processing the text input. Gray results refer to the model input resolution as 224 and the default is 448.

| | Class. | Clust. | PairClass. | Rerank. | Retr. | STS | Summ. | Avg. |
|---|---|---|---|---|---|---|---|---|
| Num. Datasets | 12 | 11 | 3 | 4 | 15 | 10 | 1 | 56 |
| Glove | 57.29 | 27.73 | 70.92 | 43.29 | 21.62 | 61.85 | 28.87 | 41.97 |
| Komninos | 57.65 | 26.57 | 72.94 | 44.75 | 21.22 | 62.47 | 30.49 | 42.06 |
| BERT | 61.66 | 30.12 | 56.33 | 43.44 | 10.59 | 54.36 | 29.82 | 38.33 |
| SimCSE-BERT-unsup | 62.5 | 29.04 | 70.33 | 46.47 | 20.29 | 74.33 | 31.15 | 45.45 |
| CLIP-T | 60.17 | 32.7 | 75.4 | 46 | 14.76 | 65.7 | 30.29 | 42.9 |
| OpenCLIP-T | 59.2 | 36.61 | 72.43 | 47.91 | 28.05 | 70.43 | 26.57 | **47.76** |
| CLIP-V | 55.76 | 31.64 | 63.85 | 45.12 | 14.51 | 62.55 | 26.81 | 40.34 |
| OpenCLIP-V | 49.4 | 23.85 | 56.55 | 42.05 | 11.75 | 54.6 | 28.57 | 34.71 |
| Im-Tx (LAION) | 49.04 | 27.67 | 67.34 | 43.67 | 16.49 | 65.26 | 29.74 | 39.27 |
| Im-Tx | 52.46 | 34.48 | 70.67 | 47.19 | 19.58 | 65.27 | 30.64 | 42.62 |
| Tx-Tx | 51.12 | 33.26 | 70.62 | 46.56 | 17.89 | 65.51 | 26.72 | 41.56 |
| IN-IN | 53.83 | 35.13 | 73.27 | 48.03 | 20.59 | 68.48 | 29.31 | 44.06 |
| Omni | 53.69 | 36.75 | 72.34 | 48.10 | 21.93 | 67.18 | 28.44 | 44.41 |
| Omni+ | 53.25 | 36.95 | 72.50 | 48.34 | 23.07 | 67.62 | 27.91 | **44.76** |
| Omni++ | 52.95 | 36.99 | 71.99 | 48.29 | 22.27 | 67.58 | 27.79 | 44.45 |

all text into images and use the pooled representation as text embedding. We can observe that OmniContrast achieve competitive performance against most of unsupervised baselines, including Glove (Pennington et al., 2014), Komninos (Komninos & Manandhar, 2016), BERT (Devlin, 2018) and SimCSE (Gao et al., 2021), which are trained on a large language corpus. When training with one-hop pair samples as the alignment target, our model achieves better performance. Similar to the aforementioned findings, the MTEB benchmark shows that the multi-modal data helps the model to better learn language representation from pixels. We also provide the results of the text(-T) and vision(-V) encoder performance of CLIP and OpenCLIP, where the vision encoder input is rendered text at 224 resolution size. Interestingly, the text encoder of OpenCLIP outperforms all the unsupervised baselines while its vision encoder poorly understands the visual text information.

## 6 DISCUSSION: WHY UNIFYING IN PIXELS?

**Motivation.** In real-world scenarios, much of image-text interleaved content is natively present in visual formats such as screenshots. Therefore, it is natural to develop a single end-to-end modal that can process any modality. Unifying everything into pixels can reduce specialized design for diverse modalities. Moreover, CLIPPO (Tschannen et al., 2023) demonstrates that the vision encoder can learn meaningful textual representation directly from pixels. While OmniContrast taking a further step towards a more general-purpose vision-centric encoder that can seamlessly understand image, scene text, and their relationship. We acknowledge that layout information (size and position) of image-text can be one major benefit of unified pixel space, which has not been fully explored in OmniContrast. Because it requires acquiring the exact snippet location from screenshots and is non-trivial to manipulate the data content, which we left for future work.

**Separate Encoder Baseline.** Besides unifying in pixel space, another straightforward approach to training CLIP on image-text interleaved data is fusing the image-text in the feature space, similar

**Table 6:** AnyCIR benchmark with Separate Encoder Baselines.

| Model | Data | IN-IN | IN-Tx | IN-Im | Tx-IN | Tx-Tx | Tx-Im | Im-IN | Im-Tx | Im-Im | Overall |
|---|---|---|---|---|---|---|---|---|---|---|---|
| OpenCLIP-V+T (B/16) | LAION 2B | 43.38 | 39.29 | 28.32 | 38.58 | 35.27 | 19.65 | 28.57 | 19.95 | 23.84 | 30.76 |
| CLIP-V+T (L/14) | WIT 400M | 43.62 | 38.72 | **28.74** | **37.97** | 33.06 | **21.30** | **28.99** | **20.41** | **23.66** | **30.72** |
| UniIR-CLIP (L/14) | UniIR-1M | **48.76** | **41.13** | 27.61 | 35.54 | **41.23** | 12.89 | 27.43 | 6.68 | 22.58 | 29.31 |
| CLIP-V+T (B/16) | WIT 400M | 37.35 | 33.18 | **24.88** | 32.59 | 28.29 | **15.92** | **24.46** | 14.40 | **21.05** | 25.79 |
| Omni (B/16) | MMC4-core | **78.27** | **73.89** | 22.10 | **74.19** | **74.32** | 10.08 | 22.00 | 10.95 | 19.50 | **42.81** |

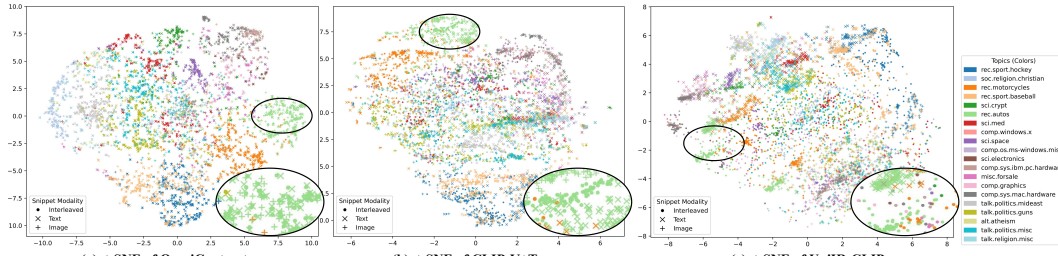

(a). t-SNE of OmniContrast      (b). t-SNE of CLIP-V+T      (c). t-SNE of UniIR-CLIP

**Figure 5:** t-SNE visualization of interleaved, text and image snippets embedding on OBELICS.

to UniIR (Wei et al., 2023). In Table 6, we report the CLIP-V+T and OpenCLIP-V+T baselines, which use feature averaging to represent image-text interleaved modalities, on our proposed AnyCIR benchmark. Moreover, we include the UniIR fine-tuned CLIP score fusion model result as the model is fine-tuned on diverse data including image-text interleaved document snippets. It can be observed that using a consistent performance drop on image-related retrieval tasks of OmniContrast after training on image-text interleaved data is the same as the UniIR trained on diverse data. The reason might be that loose image-text correspondence decreases the model capacity in image perception. Image-caption and image-text interleaved data mixing strategy can be a promising solution for this issue, we also leave this direction for future exploration.

**Benefits from Unified Pixels Space.** In Fig. 5, we visualize the distribution of interleaved, image and text embeddings from the same snippets of three models including OmniContrast, CLIP-V+T, and UniIR-CLIP. The labels of the snippet are predicted by topic model (Grootendorst, 2022) trained on 20NewsGroups (Lang, 1995). It can be observed that our model can learn useful representations that are aligned with linguistic semantics as snippets on similar topics are close to each other. Compared to the separate encoder baselines, OmniContrast learn a more unified omni-modality representation, which indicates unifying in pixel space can further reduce the modality discrepancy.

## 6.1 ABLATION STUDY AND VISUALIZATION

**Effect of Model Initialization.** As shown in Table 7a, we observed that the CLIP initialization is important for OmniContrast. Note that our training data only contains 5 million documents with around 17 million images, which is relatively small compared to WIT-400M. The scale-up experiments are left for future study due to the computation constraint and limited data scale.

**Importance of Image Rendering Positions.** In Table 7b, we ablate the effect of the image rendering position in girds as text content uses a fixed rendering order. We rendered all the image content into the same grid positions for queries, while the candidates still use random positions. The results indicate that OmniContrast learns a robust representation against different rendered grid positions.

**Modality Masking Ratio Selection.** In Table 7c, we investigate the modality masking ratio of training data. It can be observed that modality masking is crucial for image-to-image retrieval ability learning. In our setting, the best masking ratio is 40% and the larger ratio will drop the performance.

**Effect of Text Masking.** Table 7d reports the results of applying different text masking ratios during training. We find that randomly dropping the sentences in the text can improve the performance of language understanding. One possible reason is that the longer text has more redundant information.

**Non-Consecutive Pair Sampling.** As shown in Table 7e, we compare models using different ratios of one-hot consecutive pair for training. Generally, more consecutive pairs achieve higher performance on the AnyRIC benchmark as these data are more aligned with AnyRIC tasks. The one-hop consecutive pairs only slightly degrade the performance, which indicates that the model can learn useful representation from the non-consecutive snippets with a weaker connection.

**Table 7:** Ablation experiments on AnyCIR benchmark

| Init | Model | IN-IN | Tx-Tx | Im-Im | Avg |
|---|---|---|---|---|---|
| | IN-IN | 65.85 | 64.55 | 6.46 | 29.60 |
| ✓ | IN-IN | **76.56** | **74.81** | **8.95** | **34.79** |
| | Omni | 62.30 | 61.22 | 12.18 | 30.42 |
| ✓ | Omni | **78.27** | **74.32** | **19.50** | **42.81** |

**(a)** Model initialization.

| Position | Im-IN | Im-Tx | Im-Im |
|---|---|---|---|
| grid-0 | 22.07 | 10.88 | 19.53 |
| grid-1 | 22.18 | 11.03 | 19.50 |
| grid-2 | 22.01 | 10.91 | 19.51 |
| grid-3 | 22.18 | 11.03 | 19.43 |

**(b)** Image Rendering Positions.

| Ratio | IN-IN | Tx-Tx | Im-Im | Avg |
|---|---|---|---|---|
| 0.0 | 76.56 | **74.81** | 8.95 | 34.79 |
| 0.2 | 76.22 | 71.63 | **19.50** | 41.74 |
| 0.4 | 77.41 | 72.39 | 19.30 | **41.98** |
| 0.6 | 77.60 | 73.29 | 18.74 | 41.75 |
| 0.8 | **78.00** | 73.96 | 17.06 | 40.80 |
| 1.0 | 76.56 | 74.26 | 8.71 | 34.70 |

**(c)** Modality Masking.

| Ratio | IN-IN | Tx-Tx | Im-Im | Avg |
|---|---|---|---|---|
| 0.0 | 77.41 | 72.39 | 19.30 | 41.98 |
| 0.2 | **78.34** | 74.26 | 19.27 | 42.71 |
| 0.4 | 78.27 | **74.32** | 19.50 | **42.81** |
| 0.6 | 77.70 | 73.56 | 19.48 | 42.48 |
| 0.8 | 77.85 | 73.32 | **19.58** | 42.42 |
| 1.0 | 77.41 | 72.60 | 19.08 | 41.96 |

**(d)** Text Masking.

| Ratio | IN-IN | IN-Tx | IN-Im | Avg |
|---|---|---|---|---|
| 0 | **78.27** | **74.32** | 19.50 | **42.81** |
| 0.1 | 78.04 | 73.53 | **19.74** | 42.54 |
| 0.2 (+) | 77.94 | 73.68 | 19.29 | 42.52 |
| 0.3 | 78.13 | 73.65 | 19.31 | 42.44 |
| 0.4 (++) | 78.05 | 73.41 | 19.55 | 42.38 |
| 0.5 | 77.95 | 73.54 | 19.29 | 42.31 |

**(e)** Non-Consecutive Pair Sampling.

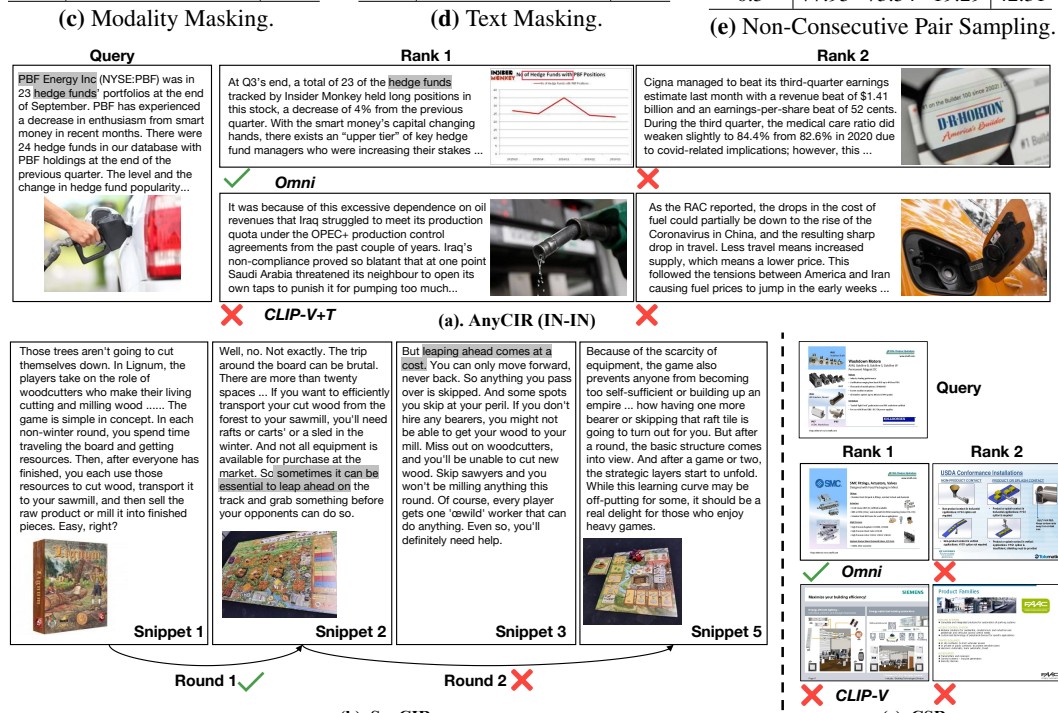

**Figure 6:** Visualization of retrieval results on AnyCIR, SeqCIR, and CSR benchmarks.

**Retrieval Results Visualization.** As shown in Fig. 6(a) OmniContrast understands the loosely vision-language correspondence correctly while CLIP-V+T is dominated by the image feature in AnyCIR IN-to-IN task. In Fig. 6(b), it can be observed that SeqCIR is a very challenging task as it requires the modal to capture the precise connection between the consecutive snippets from omni-modality input. Lastly, Fig. 6(c) indicates that despite being trained on rendered data, Omni-Contrast can effectively generalize to real-world complex layouts with different font size and style.

# 7 CONCLUSION

We introduce OmniContrast, a unified vision model that learns the loosely vision-language correspondence from multi-modal documents in a contrastive fashion. To achieve this, OmniContrast use consecutive image-text interleaved snippets as contrast targets and unify all the modalities into the pixel space. Moreover, we propose three consecutive information retrieval benchmarks to demonstrate that multi-modal web documents can empower the CLIP model with new omni-modality understanding capacity. We hope that OmniContrast serves as a stepping stone for exploring multi-modal documents as valuable training data in the vision-language research community.

Although our presented OmniContrast can process any modality input from pixel space using a single model, its efficiency and scalability are limited by its fixed input size. Future work on designing a dynamic input strategy or specific architecture could significantly enhance the performance and unlock more application scenarios for multi-modal web document understanding.

## ETHICS STATEMENT

A primary concern in our work is that the multi-modal document datasets collected from the Internet through common web crawlers may contain unfair or biased data. Despite employing multiple filtering steps during the dataset collection process, the presence of unwanted data remains a possibility. Additionally, using a pre-trained CLIP (Radford et al., 2021) checkpoint for model initialization could propagate existing biases inherent in the pre-trained model into our methodology. We are committed to continuously monitoring and mitigating potential biases in both our model and dataset as they are identified. We hope that our research contributes positively and fairly to the field of vision-language understanding research.

## REPRODUCIBILITY STATEMENT

In this work, we solely use publicly available datasets for the model training and evaluation benchmark. The CLIP (Radford et al., 2021) pre-trained model used for model initialization is fully open-source. For methodology details, we elaborate on the data preprocessing steps in Sec. 3.1 and Sec. A. Our training code base is built upon the OpenCLIP (Ilharco et al., 2021) open-source code base. Our codes and proposed evaluation benchmark data will be released upon completion of the review process.

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

## A  MORE IMPLEMENTATION DETAILS

**Data Pre-processing.** Given a document, we chunked the document into several snippets in a sliding window strategy based on text sequence. For MMC4 (Zhu et al., 2024), the document text is stored in a list of sentences. To create snippets, we merge consecutive sentences until their combined length reaches 1100 characters or less. Then we use the image-text assignment provided by MMC4 to assign each image to the corresponding snippet. For OBELICS (Laurençon et al., 2024), we first split the text content based on the newline character and then use the same sliding window strategy to generate text snippets. Differently, OBELICS organizes the documents as an image-text interleaved sequence, where the image position is extracted from the original HTML files. In both AnyCIR and SeqCIR, we assign each image to the closest preceding text snippet, while images appearing at the beginning of the document are assigned to the first text snippet.

**Training Data Details.** During training, to maintain optimal text length, we apply text masking augmentation only to snippets containing more than four sentences and exceeding 250 characters. Empirically, we found that a maximum text length of 768 characters during training led to better performance. During testing, the model can handle up to 1,100 characters without any degradation in performance. Therefore, we set the maximum training text length to 768 characters and 1,100 characters for testing. After initialization from the CLIP pre-trained checkpoint, the positional embedding is randomly initiated for $448 \times 448$ input size. For each training batch, the data modalities are mixed from image, text, and image-text interleaved without specialized balance.

## B  ADDITIONAL EXPERIMENT ANALYSIS

Table 8 presents the complete results of the AnyCIR benchmark used in the ablation study. Table 9 shows the ablation study on padding text to exceed a certain length by repeating it and its impact on M-BEIR task performance. The results suggest that the short text information might be surpassed in the image-text interleaved representation.

## C  VISUALIZATION

In Fig. 7, we showcase some rendered snippet samples used for training. Moreover, we present some examples of our proposed consecutive information retrieval benchmark, shown in Fig. 8,9 and 10.

**Table 8:** Full results of ablation study in AnyCIR.

| Settings | | IN-IN | IN-Tx | IN-Im | Tx-IN | Tx-Tx | Tx-Im | Im-IN | Im-Tx | Im-Im | Overall |
|---|---|---|---|---|---|---|---|---|---|---|---|
| - | IN-IN | 65.85 | 64.26 | 0.10 | 63.84 | 64.55 | 0.05 | 1.10 | 0.19 | 6.46 | 29.60 |
| Init ✓ | IN-IN | 76.56 | 74.85 | 0.40 | 74.19 | 74.81 | 0.12 | 2.58 | 0.64 | 8.95 | 34.79 |
| - | Omni | 62.30 | 59.29 | 8.52 | 59.11 | 61.22 | 1.47 | 8.23 | 1.49 | 12.18 | 30.42 |
| Init ✓ | Omni | 78.27 | 73.89 | 22.10 | 74.19 | 74.32 | 10.08 | 22.00 | 10.95 | 19.50 | 42.81 |
| Image Rendering Positions | grid-0 | 78.17 | 73.96 | 22.15 | 74.38 | 74.32 | 10.12 | 22.07 | 10.88 | 19.53 | 42.84 |
| | grid-1 | 78.26 | 74.05 | 22.07 | 74.38 | 74.32 | 10.12 | 22.18 | 11.03 | 19.50 | 42.88 |
| | grid-2 | 78.31 | 74.01 | 22.00 | 74.38 | 74.32 | 10.12 | 22.01 | 10.91 | 19.51 | 42.84 |
| | grid-3 | 78.18 | 73.78 | 22.04 | 74.38 | 74.32 | 10.12 | 22.18 | 11.03 | 19.43 | 42.83 |
| Modality Masking Ratio | 0.0 | 76.56 | 74.85 | 0.40 | 74.19 | 74.81 | 0.12 | 2.58 | 0.64 | 8.95 | 34.79 |
| | 0.2 | 76.22 | 71.47 | 21.94 | 71.44 | 71.63 | 10.67 | 21.56 | 11.25 | 19.50 | 41.74 |
| | 0.4 | 77.41 | 72.06 | 21.72 | 72.74 | 72.39 | 9.71 | 21.78 | 10.72 | 19.30 | 41.98 |
| | 0.6 | 77.60 | 73.35 | 20.72 | 72.90 | 73.29 | 9.02 | 20.70 | 9.47 | 18.74 | 41.75 |
| | 0.8 | 78.00 | 74.32 | 17.38 | 73.93 | 73.96 | 6.89 | 17.96 | 7.69 | 17.06 | 40.80 |
| | 1.0 | 76.56 | 74.49 | 0.54 | 74.07 | 74.26 | 0.26 | 2.78 | 0.65 | 8.71 | 34.70 |
| Text Masking Ratio | 0.0 | 77.41 | 72.06 | 21.72 | 72.74 | 72.39 | 9.71 | 21.78 | 10.72 | 19.30 | 41.98 |
| | 0.2 | 78.34 | 73.96 | 21.85 | 74.25 | 74.26 | 10.16 | 21.46 | 10.89 | 19.27 | 42.71 |
| | 0.4 | 78.27 | 73.89 | 22.10 | 74.19 | 74.32 | 10.08 | 22.00 | 10.95 | 19.50 | 42.81 |
| | 0.6 | 77.70 | 73.44 | 21.94 | 73.42 | 73.56 | 10.11 | 21.88 | 10.77 | 19.48 | 42.48 |
| | 0.8 | 77.85 | 73.20 | 21.86 | 73.20 | 73.32 | 10.11 | 22.01 | 10.64 | 19.58 | 42.42 |
| | 1.0 | 77.41 | 72.38 | 21.60 | 72.66 | 72.60 | 9.67 | 21.64 | 10.61 | 19.08 | 41.96 |
| Consecutive Pair Sampling | 0.0 | 78.27 | 73.89 | 22.10 | 74.19 | 74.32 | 10.08 | 22.00 | 10.95 | 19.50 | 42.81 |
| | 0.1 | 78.04 | 73.27 | 21.88 | 73.66 | 73.53 | 9.90 | 21.96 | 10.94 | 19.74 | 42.54 |
| | 0.2 | 77.94 | 73.68 | 21.87 | 73.73 | 73.68 | 10.06 | 21.76 | 10.70 | 19.29 | 42.52 |
| | 0.3 | 78.13 | 73.46 | 21.46 | 73.76 | 73.65 | 9.98 | 21.51 | 10.68 | 19.31 | 42.44 |
| | 0.4 | 78.05 | 73.53 | 21.27 | 73.57 | 73.41 | 9.96 | 21.48 | 10.63 | 19.55 | 42.38 |
| | 0.5 | 77.95 | 73.50 | 21.29 | 73.37 | 73.54 | 9.80 | 21.59 | 10.47 | 19.29 | 42.31 |

**Table 9:** Ablation study of text padding length on M-BEIR benchmark.

| Task | Dataset | Text Padding Length | | | | |
|---|---|---|---|---|---|---|
| | | - | 100 | 400 | 800 | 1000 |
| $(q_i, q_t) \rightarrow (c_i, c_t)$ | oven_task8 | 0.26 | 0.65 | 4.37 | 5.77 | 5.21 |
| | infoseek_task8 | 0.09 | 0.33 | 3.01 | 4.21 | 4.05 |

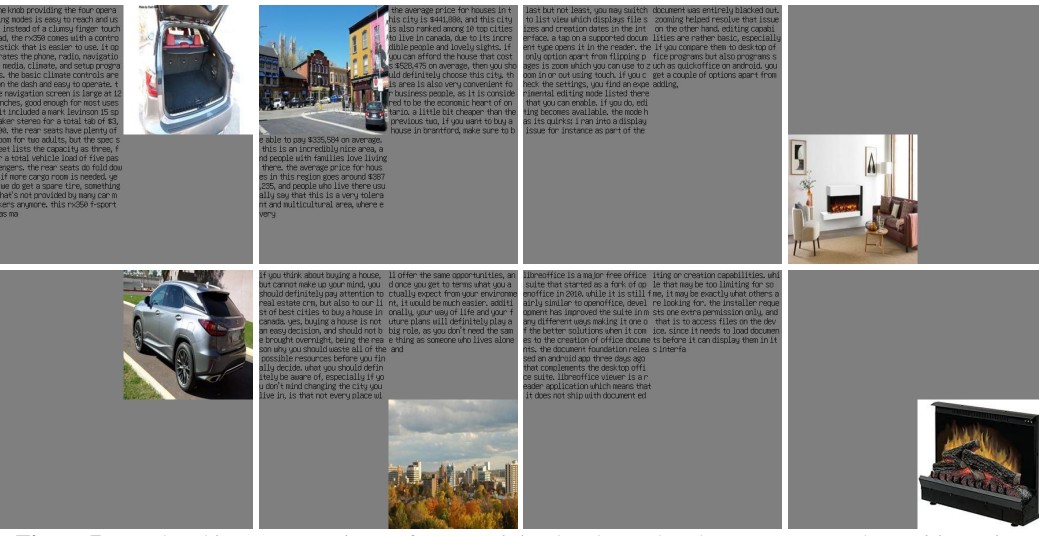

**Figure 7:** Rendered image-text snippets from a training batch. Each column represents the positive pairs.

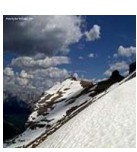

I learned a lot while researching this article, with the help of conversations with Mike Brown, Pablo Santos-Sanz, and Alex Parker. I did the original research for it nearly two years ago now, when I wrote this post about the shapes of Kuiper belt orbits, and I want to thank Mike especially for a recent review to make sure it was still up-to-date, and Alex for helping me figure out the colors of the largest Kuiper belt objects. While I'm mentioning magazines, I want to congratulate amateur image processor Gordan Ugarkovic for his version of Cassini's top-down Saturn portrait making the cover of the January/February 2014 issue of Discover magazine. In the writeup about the cover image, photo editor Ernie Mastroianni wrote:

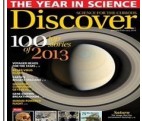

[Ugarkovic] posted the final stunning image to an online forum run by the non-profit Planetary Society, an organization that promotes space exploration. The photograph went viral after senior editor and planetary evangelist Emily Lakdawalla posted it on her blog and Twitter. At Discover, we were so impressed that we judged it our favorite science image of 2013 and placed it on our cover. The amateur image processing community helps NASA, ESA, and other space agencies put their best foot forward -- thanks, Gordan, and the folks at unmannedspaceflight.com, for making this happen! Discover: The Year in Science 2013 I've been waiting impatiently for this issue to show up in my mailbox since seeing your name on the cover -- and it just arrived today! Can't wait to read the article.

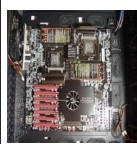

There are four alpine routes up Castle Mountain listed in Sean Dougherty's "Selected Alpine Climbs in the Canadian Rockies". Eisenhower Tower, Bass Buttress, Ultra-Brewers and Brewer Buttress. Quite a few other routes can be contemplated at Tabvar.org. Bass Buttress, Brewer Buttress and Eisenhower are the "classics" and therefore most common routes. What makes Bass Buttress popular no doubt is the access via the tiny Castle Mountain Hut (photo provided) managed by the Alpine Club of Canada. Don't have any grand illusions of throwing a party up there. Although advertised to sleep six, I feel sorry for the last two of six to arrive. It is a very cool location for a hut though and even though you can do Bass Buttress easy in a day from the car (as we did), the hut is an experience in and of itself not to miss. And perhaps even more unique is the open air pit toilet on the edge of a dramatic cliff.

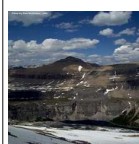

Bass Buttress was put in by Brian Greenwood and Joe Farrand in 1968. Bass Buttress Direct, the version I did, was put up by Bugs McKeith and John Calvert in 1972 and I much recommend this line over the original, which involves three alternate pitches raising the rating from 5.6 to 5.7. It is a shaded route for much of the day, which is a huge advantage on hot summer days, but at this elevation, we are only talking a few days of the year that this would be seen as an advantage. Bass Buttress has less pins than Brewers Buttress and is climbed somewhat less because it normally is considerably colder. The direct route makes it a much cleaner line. This is a 4600'+/- total ascent trip, car to car. The guidebook discusses some 3.5-5 hours to achieve the hut from the parking area via the Castle Lookout Trail. However I typically take only 2 hours.

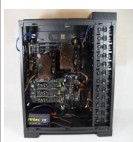

There are many trick components to this build, but one of the most over the top was sent to me from G.Skill. G.Skill released a very special memory kit for the SR-2 based on the Trident heat spreader design. This kit was only available in 48GB and 24GB capacity sizes and had an official rating of 2,000 MHz with 9-8-9-24 timings. That didn't stop me from starting out at 2K MHz and 7-7-7-20 timings. That's good enough for this old timer. When the system gets settled in I'll make some forum posts on this very fast memory. A big thank you goes out to all of the companies who thought it was time for me to get back into kicking virtual ass. My build started out simple enough. The Xigmatek Elysium comes with enough motherboard stand offs to install the EVGA SR-2. What's even more impressive is the amount of usable cable push thru locations left even after putting in this massive board. We still have two at the top, two at the bottom and four on the drive bay side that can use utilized.

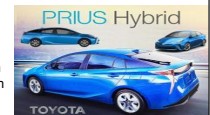

The back side will allow me to quickly install a water cooling kit when I start feeling the urge to overclock the six-core Xeons. The system is starting to come together and I found a way to route the USB 3.0 cables. At first I didn't think these were long enough to reach the rear USB 3.0 ports on the motherboard, but they made it with enough slack to not be concerned about. This is also one of the things I didn't really like about the Elysium and something I hope Xigmatek changes in future versions. New motherboards are shipping with internal USB 3.0 headers, but the case doesn't come with adapters to switch from external connectors and the new USB 3.0 headers. This was brought up with Xigmatek and they are taking it under consideration. Go big or go home baby! Three way SLI with dual Xigmatek Hammers cooled by 120mm fans that match the included rear and HDD bay fans. In the future I'll need to brush up on my cable management skills or just force Chad to do it when he comes to visit.

**Figure 8:** Visualization samples in AnyCIR benchmark. Each row represents the consecutive pairs.

Round 1

What is a Toyota Prius hybrid vehicle? Prius is the world's first mass-produced hybrid model launched by Toyota Motor in Japan in 1997. In 2001, it has been sold to more than 40 countries and regions around the world, and its main markets are Japan and North America. Among them, the United States is the largest market for Prius. As of the beginning of 2009, Prius has sold more than 600,000 vehicles in the United States. According to 2007 data from the U.S. Environmental Protection Agency, the Prius is the most fuel-efficient car sold in the United States. Additionally, the Prius is by far the cleanest vehicle in the United States, according to the U.S. Environmental Protection Agency and the California Air Resources Board's evaluation of each model based on carbon dioxide emissions. According to figures released by the UK Department for Transport, the Prius is the second-lowest CO2-emitting vehicle sold in the UK.

Round 2

The first-generation Prius came out at the end of October 1997 and was the world's first mass-produced hybrid vehicle. Today, when people pay more and more attention to environmental protection, Prius has epoch-making significance and advancement because of its revolutionary reduction of vehicle fuel consumption and exhaust emissions, and has been highly praised by the world. The shape of the Prius is shown in Figure 1. (1) As needed, the engine can be stopped and the motor can be driven alone. Regardless of starting or normal driving, the electric motor will be preferentially driven, thereby shortening the working time of the engine and achieving the goals of low fuel consumption, low emissions and low noise.

Round 3

(2) Fully recover the energy and charge the battery to effectively reuse the energy. When decelerating, the engine can be completely stopped, and the wheel drives the generator to charge the battery for efficient energy recovery. The large-capacity battery can achieve more power storage. (2) Electric motor. The maximum output power of the electric motor equipped in the new-generation Prius has been increased from the original 50kW to 60kW, and through measures such as increasing torque and adopting a reduction gear, it has achieved miniaturization and light weight, and further improved the fuel economy of the vehicle. (4) Power distribution device. The power of the engine is sent to the wheels and the generator respectively, and at the same time, by connecting and effectively controlling the engine, motor and generator, the vehicle has agile and smooth acceleration performance.

Round 4

(5) Battery. The new-generation Prius uses high-power nickel-metal hydride batteries, which can provide sufficient power for the motor and generator, and greatly reduce the dead area of the battery, improving energy efficiency. The cooling system and main relay are arranged in an optimal distribution way, and the air inlet and outlet of the cooling system and the fan are miniaturized, which not only brings low fuel consumption, but also reduces the body weight and expands the trunk space.

(6) Variable voltage control system. The system can effectively control the DC output of the battery and the AC output used to drive the motor and generator. The new generation of Prius can increase the system voltage from the maximum 500V of the previous generation model to 650V with the help of the boost converter of the variable voltage control system, and further optimize the cooling device, greatly improve the motor torque, and make the system smaller, lighter in weight, more efficient in operation, and more powerful in output power.

**Figure 9:** Visualization sample in SeqCIR benchmark.

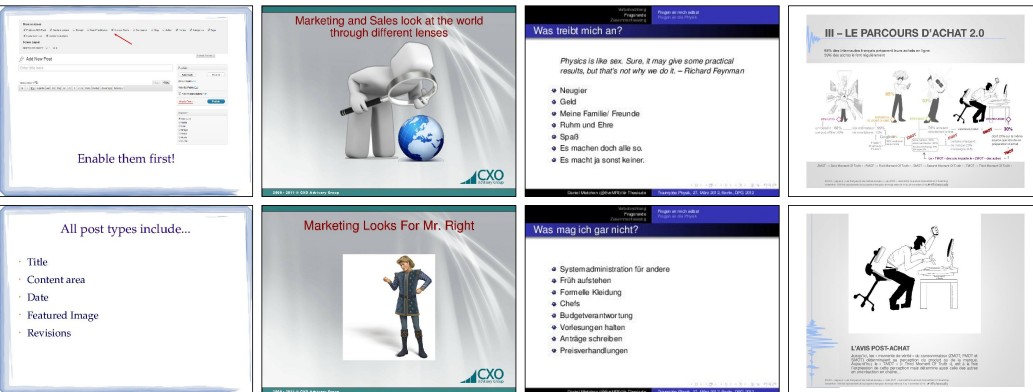

**Figure 10:** Visualization samples in CSR benchmark. Each column represents the consecutive pairs.

