# OpenReview forum: "OmniContrast: Vision-Language-Interleaved Contrast from Pixels All at once"
_ICLR.cc/2025/Conference — Submitted to ICLR 2025_

### Official Review · Reviewer_Bmsk · 2024-10-24

**Soundness:** 3
**Presentation:** 2
**Contribution:** 3
**Rating:** 6
**Confidence:** 4

**Summary:**

OmniContrast, a unified contrastive learning model for understanding vision, language, and vision-language interactions within multi-modal web documents. Unlike traditional models, OmniContrast:

- Explores a new contrastive approach to maximize similarity between consecutive snippets from image-text interleaved web documents.
- Unifies all modalities (text, images) into pixel space, rendering text visually, simplifying processing and representation.
- Enables a single vision model to process any modality.

**Strengths:**

1. Excellent ablation study demonstrating the necessity of including each modality in the proposed pipeline (Table 1).
2. Clearly outperforms baseline methods, allowing the model to work in different modality settings.

**Weaknesses:**

1. Despite the proposed method outperforming CLIPPO in terms of average scores, it seems that the baseline method is capable of handling all modalities in OmniContrast. Clarification on the contribution is needed.

2. Data augmentation of the training data is a crucial part of the pipeline, but it is not well-documented, raising concerns about synthesizing low-quality training samples.

3. Figure 2: The images and fonts are extremely small, making it difficult to understand. The caption fonts also appear too small.

4. The concept of omni-modality seems odd from a reading perspective, as it appears the authors are solving vision-language problems.

5. In the abstract, "OmniContrast unifies all modalities into pixel space, where text is rendered visually" was difficult to understand until reading the entire introduction and related work section. The term "rendering" suggests high-resolution 3D scenes, whereas simple text copying and pasting is not truly rendering.

**Questions:**

1. Does training a model in this omni-style make it easier or harder to converge?
2. Related to Q1, do the authors believe that adding modalities helps the model learn each modality better, or does it make the training problem more complicated?
3. What would happen to OmniContrast if there were abundant data in three modalities but limited data in the fourth modality?

---

> ### Author Response · Authors · 2024-11-20
>
> Thank you for your detailed observations and questions. Below, we provide clarifications and propose adjustments to enhance the manuscript based on your valuable feedback.
>
> **Contribution:**
>
> We sincerely request the reviewer to re-evaluate our contribution from two perspectives:
> - **New Training Fashion:**
> As highlighted by Reviewer \#naQ8, we are the `first to explore the potential of image-text interleaving documents` for training CLIP-style models, which has been underscored in the vision-language research community.
> - **More Unified Model:**
> Our approach is capable of `directly handling long text and images with embedded text` in a unified model, as evidenced by Table 5 and Table 3—capabilities that CLIPPO or other baseline models lack.
> In Section 6, we showcase that OmniContrast learns a more unified omni-modality representation, which indicates unifying in pixel space can further reduce the modality discrepancy.
>
>
> **Data Augmentation:**
>
> We understand the importance of clearly explaining our data augmentation strategies to address concerns about training sample quality. Our augmentation techniques include modality masking and text masking, designed to enhance diversity and robustness during training.
> - *Modality masking* involves randomly dropping one modality from image-text interleaved samples, such as removing the text while retaining only the image.
> - *Text masking* randomly removes sentences from the beginning or end of the text content when the text contains more than four sentences and exceeds 250 characters.
>
> Detailed descriptions of these strategies are provided in the supplementary material.
> The effectiveness of these augmentations is demonstrated through ablation studies in Section 6.1, which show that they not only increase training diversity but also improve the model’s robustness to unseen inputs. Additionally, we include visualizations of our training data in the supplementary material to further illustrate these techniques.
>
>
> **Figure2 \& Terminologies:**
>
> Thanks for the suggestion!
> - We will improve the readability of Figure 2.
> - We will revise the manuscript to clarify that ``omni-modality" refers to an image, text, and image-text interleaving.
> This term will be explained early in the paper to avoid confusion.
> - We will add a footnote clarifying that here it refers to visually presenting text, akin to copying and pasting onto an image, to avoid confusion.
>
>
> **Converge and Benefits from Omni-style:**
>
> Based on the training loss curves, we observed the training in omni-style makes it harder to converge than in image-text style.
> However, our experiment results, e.g. Table 1, have shown that training a vision encoder with omni-modality data generally helps each modality learn better.
>
>
> **Limited data:**
>
> The training data in OmniContrast is imbalanced across different modalities. Specifically, the proportion of image-to-image pairs is smaller than that of text-to-text pairs due to the limited number of images compared to the larger volume of text chunks. Consequently, as shown in Table 2, the performance on the image-to-image task is lower than that of the text-to-text counterpart.

---

> > ### Comment · Reviewer_Bmsk · 2024-11-27
> >
> > Thanks for the response. I continue to believe it's a valid work with limitations.
> >
> > I will keep my rating. I wish there is the score 7 option.

---

### Official Review · Reviewer_MsPN · 2024-11-02

**Soundness:** 3
**Presentation:** 3
**Contribution:** 3
**Rating:** 6
**Confidence:** 2

**Summary:**

This paper presents the OmniContrast model, which unifies vision and text into a pixel space for web document retrieval and understanding. Moreover, this paper presents three new information retrieval benchmarks (AnyCIR, SeqCIR, and CSR) to evaluate the ability of the model to retrieve continuous information in complex multi-modal documents.

**Strengths:**

- The model performs excellently, achieving outstanding results in multiple baselines.
- Good writing and detailed experiments.
- A novel and useful approach for transforming interleaved data into pixel space.

**Weaknesses:**

- I'm not sure if I'm misunderstanding the model, but I think there is a lack of comparisons on some baselines, such as VQAv2 and GLEU like the comparisions in CLIPPO.
- I think there is a lack of further discussion on the necessity and effectiveness of unifying text and images into pixel space, as well as a comparison of the differences between interleaved data and text-image pairs in this unified pixel space.

**Questions:**

I believe that the handling of interleaved data is a significant distinction between OmniContrast and CLIPPO.

Therefore, I'm curious about the differences in the model's performance when using interleaved data compared to image-text pairs.

---

> ### Author Response · Authors · 2024-11-20
>
> Thanks for the constructive feedback! We hope to clarify the confusion and answer the questions below.
>
> **Comparison Baselines:**
>
> *Our research emphasizes the importance of image-text interleaving and long-text scenarios.*
> *While traditional benchmarks like VQAv2 and GLUE tend to focus on short sentences and have become somewhat outdated.*
> Therefore, we have selected two more comprehensive benchmarks: M-BEIR, which offers more diverse text-image retrieval settings, and MTEB, which provides more extensive language understanding scenarios with longer text.
> Additionally, the more complex QA setting and some tasks from GLUE are included in these well-developed benchmarks (Line 324-354, Section 5.3 and 5.4).
> By integrating these diverse benchmarks, we offer a more comprehensive evaluation result.
> We respectfully invite the reviewer to carefully examine these results to gain a deeper understanding of our work.
>
> **Necessity and Effectiveness of Unifying:**
>
> - **Necessity:** We would like to emphasize that unifying everything into pixels can `avoid specialized design` for diverse modalities, which significantly `reduces the complexity` of the training and inference pipeline.
> Moreover, our approach is capable of directly handling images with embedded text and long text, as evidenced by Table 3 and Table 5—capabilities that baseline models lack.
> - **Effectiveness:** Extensive experiments in Table 6 show that OmniContrast surpasses the separate encoder in terms of `smaller training data scales and model sizes.`
> We showcase that OmniContrast learns a more unified omni-modality representation, which indicates unifying in pixel space can further reduce the modality discrepancy.
> In Section 6, we provide a detailed discussion of the necessity and effectiveness of unifying pixels.
>
>
> **Comparison on image-text paired data:**
>
> In our experiments, *we have included an image-text data baseline trained on the LAION 40M subset for comparison, i.e., Im-Tx baseline trained on LAION-40M (L276 in Table 1, L289 in Table 2\&3, Table 4 Im-Tx$_{la}$, L349 Im-Tx (LAION) in Table 5).*
> Several notable differences emerged between our approach and the image-text paired baseline.
> - For instance, as shown in Table 5, the image-text baseline performs better on the NIGHTS [1] retrieval dataset, where the task requires retrieving identical images. This suggests that the image-text paired baseline is more effective at capturing fine-grained image details.
> - However, in terms of text embedding quality, as presented in Table 6, the image-text paired baseline falls significantly behind Omni-Contrast, indicating that our approach excels in text representation as the model is trained with longer text.
>
> [1] Fu, Stephanie, et al. Dreamsim: Learning new dimensions of human visual similarity using synthetic data. arXiv:2306.09344.

---

### Official Review · Reviewer_JeXw · 2024-11-02

**Soundness:** 2
**Presentation:** 3
**Contribution:** 3
**Rating:** 5
**Confidence:** 3

**Summary:**

This paper develops OmniContrast to unify vision-language modeling from image-text interleaved web data. To evaluate such a unified model, the authors develop the AnyCIR and SeqCIR benchmarks. These two benchmarks focus on evaluating the relevant snippet retrieval ability of the model.

**Strengths:**

- Clear presentation.

- The evaluation of different methods on AnyCIR and SeqCIR seems sound.

- The method is also straightforward, only a unified model saves the memory.

**Weaknesses:**

- The reviewer appreciates the development of benchmarks like AnyCIR and SeqCIR. One pitty is that the results of baselines are all reproduced by the authors. No third-party baselines are provided.

- No results on common benchmarks are provided. In this case, the reviewer may think that OmniContrast is only developed for CIR, this specific task. It may discount the contribution of this work.

**Questions:**

- In Section 5.2, do the authors only use the vision encoder of CLIP/OpenCLIP for evaluation? Why not use the full CLIP/OpenCLIp model?

- Could the authors provide results on common benchmarks like MS-COCO (text-to-image retrieval), Flickr30k (text-to-image retrieval), and GLUE benchmark? Like what CLIPPO [1] did. The reviewer thinks this can better figure out what can/cannot OmniContrast do.
    - As said in the  Weaknesses, all results of baselines are reproduced by the authors. Comparisons on common benchmarks make the evaluation more strong.

- Another question is, why we would choose OmniContrast when there are many next-token-prediction VLMs? For example, the Emu series[2]. Such VLMs may be the mainstream now. The reviewer thinks these VLMs can also do what OmniContrast can do. Relevant discussions/comparisons are required.


[1] https://arxiv.org/pdf/2212.08045

[2] https://github.com/baaivision/Emu

---

> ### Author Response · Authors · 2024-11-20
>
> **About third-party baselines:** *`It is not true.`*
>
> We do include third-party baselines `in Table 4 (M-BEIR) and Table 5 (MTEB)`.
> There are several baselines included from third parties, such as SigLIP (Table 4), BLIP (Table 4), BLIP2 (Table 4), Glove (Table 5), Komninos (Table 5), BERT (Table 5), and SimCSE-BERT-unsup  (Table 5).
>
>
> **Common Benchmarks \& Q2:** *`It is also not true.`*
>
> Our evaluations have included two common benchmarks: `M-BEIR (Table 4) and MTEB (Table 5).`
> The comparison is presented in Line 324-Line 354.
> *OmniContrast focuses on image-text interleaving and long-text scenarios.*
> *While these traditional image-text retrieval benchmarks and GLUE typically are short sentences and are well-explored.*
> Therefore, to better evaluate the capability of image-text interleaving and long-text understanding, we chose the M-BEIR (more diverse text-image retrieval settings) and MTEB (more long text settings).
> Moreover, the image-text retrieval setting and some tasks from GLUE are included in these well-developed benchmarks.
> We believe these benchmarks collectively offer a more comprehensive evaluation.
> We kindly encourage the reviewer to refer to these results to re-evaluate our work.
>
> **About Section 5.2:**
>
> Yes, in section 5.2 we only use the vision encoder of CLIP / OpenCLIP for fair evaluation.
> In Section 6.1 we provide the result of the full CLIP / OpenCLIp model for further discussion.
>
> **Discussions about Multi-Modal Large Language Models:**
>
> Thank you for the suggestion.
> Multi-modal Large Language Models (MLLMs) are not specifically tailored for retrieval tasks.
> Their primary strength lies in handling interleaved inputs for generative tasks, such as image captioning and question answering with both visual and textual inputs.
> *For example, in Table 4 M-BEIR benchmark, BLIP2[1] is similar to Emu[2] powered by the large language model and shows very limited performance on various retrieval settings.*
> Applying MLLMs to retrieval tasks [3] is another promising research problem, `but is not our focus.`
> We will discuss these methods in Section 6 of the revised paper to address this perspective.
>
> [1] Li, Junnan, et al. BLIP-2: Bootstrapping Language-Image Pre-training with Frozen Image Encoders and Large Language Models. ICML 2023
>
> [2] Sun, Quan, et al. Emu: Generative Pretraining in Multimodality. ICLR 2023
>
> [3] BehnamGhader, Parishad, et al. LLM2Vec: Large Language Models Are Secretly Powerful Text Encoders. COLM 2024

---

### Official Review · Reviewer_naQ8 · 2024-11-03

**Soundness:** 2
**Presentation:** 3
**Contribution:** 2
**Rating:** 5
**Confidence:** 4

**Summary:**

This paper proposed OmniContrast, a unified contrastive learning model that processes multi-modal web documents by transforming all modalities, including text, into pixel space for a single vision model to handle. It achieves competitive or superior performance compared to standard CLIP models, demonstrating the value of multi-modal web data for advancing vision-language learning.

**Strengths:**

1. OmniContrast is among first to explore vision-language correspondence on image-text interleaved web documents in CLIP-style.
2. Authors propose three consecutive information retrieval benchmarks, including AnyCIR, SeqCIR, and CSR to o facilitate the evaluation of omni-modality understanding.
3. The effectiveness is validated by experimental results.

**Weaknesses:**

I am concerned about the motivation with the single modality in the pixel space. I believe it is limited in a few ways.

1. It is ture that "image-text interleaved content is natively present in visual formats such as screenshots". Screenshot is a scenario, however, in more cases, such as the very rich html format image-text interleaved data (much richer than screenshots), images and texts are naturally presented in different modalities.

2. Is it really practical unifying them into pixels? In many cases, we have seperated texts and images, where we have to re-organize them in the form of "screenshots" to use the model. It can be redundant. And organizing them in the form of "screenshots" itself can involve some issues, such as the limitation from the resolution, etc. I agree that CLIPPO (Tschannen et al., 2023) demonstrates that the vision encoder can learn meaningful textual representation directly from pixels, however, "it is feasible to do so" does not mean it is a good solution in different scenarios. I am looking for a strong motivation to do so.

3. In Tab. 6, simple alternatives like CLIP-V+T, and UniIR-CLIP are very effective when compared to Omni. That is also why I am considering if unifying them into pixels is a good solution and well-motivated.

**Questions:**

### Reply (Post Rebuttal)

I do not think my comments have the inconsistencies mentioned by the authors.

> You correctly acknowledge that unifying information into a single modality simplifies the model structure and improves handling image-text interleaving data (e.g., screenshots).

These are two separate points. The authors mention two advantages: the first is simplifying the structure, and the second is the use case for screenshots. I acknowledge its usefulness for screenshots but do not consider "simplifying the structure" to be a clear benefit. These two points are entirely unrelated, so the inconsistencies claimed by the authors do not exist.

1. The reason I don’t view "simplifying the structure" as a clear benefit has already been explained: *"The text encoder in CLIP is also quite simple, and I feel this is more of a design choice between single-tower and two-tower architectures rather than a significant advantage."* While a single-tower model does eliminate the text encoder in a two-tower architecture, does removing a CLIP text encoder offer any clear advantage in most scenarios? That is the question I raised. We all know that removing a text encoder reduces the number of parameters, but if this is being presented as a major contribution and clear advantage, the authors need to demonstrate why removing a text encoder is crucial in their application context. I did not see this importance addressed in either the paper or the rebuttal.

2. The authors argue that *"we address cases where text extraction is complex or difficult, like image-text interleaving formats like screenshots."* However, recognizing printed text from screenshots is straightforward as I know.

I remain concerned about unifying text into the pixel space, where sentences are treated as a bag of words literally. And it is more concerning when the authors emphasize long-form text, where contextual dependencies are likely more important.

---

> ### Author Response · Authors · 2024-11-20
>
> Thank you for taking the time and effort to review our work.
>
> **What Makes Single Modality Good?**
>
> - Firstly, we would like to highlight that unifying in a single modality `significantly simplifies` the model architecture by reducing the complexity of the text encoder and the need for a fusion strategy typically required in separate encoder setups as recognized by the Reviewer \#Bmsk.
> - Secondly, our model is inherently designed to `handle images with image-text interleaving content directly`. In contrast, separate encoder models require additional steps for text extraction to fully utilize both modalities.
> The interleaved data commonly appears in everyday scenarios such as documents, TV shopping broadcasts, advertisements, and more, where extracting text can often be challenging.
>
> **Application Scenarios:**
> - It is important to note that in the case of HTML, `image-only or text-only inputs are simply special cases` handled seamlessly by our model! For example, as shown in Figure 2, our model already supports such single modality input during training.
> - Moreover, as suggested in [1] handling HTML is much more `complicated than plain text` and requires a `much larger input context` while it is `very straightforward and simple by handling them in image space` using screenshots.
> - Besides, many other scenarios, such as screenshots [2], slides [3] (as demonstrated in our CSR benchmark), PDFs [4], and scene text [5], present image-text interleaved content in a primarily visual format.
> These cases pose unique challenges for `extracting text content` due to their reliance on visual input.
> Therefore, a unified approach to processing image-text interleaved data is particularly valuable.
>
> **The Redundancy of Single Modality:**
>
> We acknowledge the additional effort required to re-organize image-text input into the form of screenshots for OmniContrast.
> However, this process requires a `much lower computational cost` compared to forwarding text inputs through an additional text encoder.
> Regarding the resolution, our experiments demonstrate that a carefully chosen resolution strikes a good balance between performance and input quality.
> For instance, `our approach supports a maximum text input length of 1,100 characters (around 275 tokens), while the text input of CLIP is limited to 77 tokens.`
>
>
> **Simple Alternatives:**
> *We respectfully disagree that these simple alternatives are very effective.*
>
> - *In terms of the model scale*, CLIP-V+T (ViT-L) and UniIR are in the size of `ViT-L` while our OmniContrast is in the size of `ViT-B` and their performance is still limited, e.g., 42.81 (ours) v.s 30.72 (CLIP-V+T ViT-L) v.s. 29.31 (UniIR) in overall performance. In the size of ViT-B, our model outperforms the CLIP-V+T (ViT-B) by a large margin, e.g. 42.81 (ours) v.s 25.79 (CLIP-V+T) in overall performance.
> - *In terms of training data*, our training data MMC4-core maintains a `relatively small size, i.e., 5M,` for which we believe OmniContrast has great potential to be an effective solution when scaling up the model and training data.
>
> [1] Gur, Izzeddin., et al. Understanding HTML with Large Language Models. EMNLP 2023
>
> [2] Chen, Xingyu., et al. WebSRC: A Dataset for Web-Based Structural Reading Comprehension. EMNLP 2021
>
> [3] Tito, R., et al. Document Collection Visual Question Answering. ICDAR 2021.
>
> [4] Araujo, André, et al. Large-scale query-by-image video retrieval using bloom filters. arXiv:1604.07939
>
> [5] R, Ganz, et al. Towards Models that Can See and Read. ICCV 2023

---

> > ### Comment · Reviewer_naQ8 · 2024-12-03
> >
> > Thank you for your response. I agree that unifying information into a single modality does simplify the model structure, eliminating the need for an additional text encoder. This is a fact, but I don't see it as a clear benefit. The text encoder in CLIP is also quite simple, and I feel this is more of a design choice between single-tower and dual-tower architectures rather than a significant advantage. However, I acknowledge the examples provided in your application scenarios. The ability to directly use screenshots containing both text and images is indeed a convenient approach. This might be a useful application. However, overall, I find the contribution, especially the technical contribution to be relatively limited.
> >
> > Besides, in the long term, I still believe that unifying in pixel space is not an ideal choice, especially given the rapid advancements in language models today. Many lightweight LLMs can serve as text encoders, and their pretraining undoubtedly provides unparalleled advantages in text understanding compared to unifying in pixel space.

---

> > > ### Author Response · Authors · 2024-12-03
> > >
> > > We must address some `key inconsistencies in your comments`.
> > >
> > > **Simplification is a Strength, Not a Dismissible Choice:**
> > > You correctly acknowledge that unifying information into a single modality simplifies the model structure and improves handling image-text interleaving data (e.g., screenshots). However, `you dismiss this as merely a "design choice".` This downplays the clear advantages of reduced model complexity, which leads to better development efficiency and memory savings. These are substantial benefits, not just a superficial consideration. For example, many previous works focus on unified models [1-3], while our work provides a new insight into unified modeling from pixel space.
> > >
> > > **Pixel Space vs. LLMs (Context Matters):**
> > > You imply that lightweight LLMs are the ideal text encoders, but your argument overlooks the very context in which our approach excels.  The assumption is text is easily extractable from input, `which can not directly address our cases and many multi-modal tasks.` By unifying text and image data in pixel space, we address cases where text extraction is complex or difficult, like image-text interleaving formats like screenshots.
> > >
> > > In summary, `your feedback contradicts itself by recognizing the value of our approach in certain use cases while downplaying its technical contribution.`
> > >
> > > [1] Zhou, Luowei, et al. "Unified vision-language pre-training for image captioning and vqa." AAAI 2020.
> > >
> > > [2] Lu, Jiasen, et al. "Unified-io: A unified model for vision, language, and multi-modal tasks." ICLR 2022.
> > >
> > > [3] Bao, Hangbo, et al. "Vlmo: Unified vision-language pre-training with mixture-of-modality-experts." NeurIPS 2022.

---

### Author Response · Authors · 2024-11-28
**Kindly Requesting Your Feedback on Our Rebuttal**

Dear Reviewers,

Thank you for taking the time to review our paper and provide your valuable feedback. We have carefully addressed your concerns in our submitted rebuttal. As the rebuttal period nears its conclusion, we kindly request you review our responses and share any additional comments or suggestions. Your insights are greatly appreciated, and we are grateful for your thoughtful input.

Best Regards

---

### Meta-Review · Area_Chair_rW6c · 2024-12-20

**Metareview:**

This paper introduces OmniContrast, a unified model that processes both text and images by converting everything into pixel space for a single vision model to handle. To evaluate it, the authors create new benchmarks: AnyCIR, SeqCIR, and CSR, to test the model's ability to retrieve relevant snippets.

OmniContrast is appreciated for breaking new ground by exploring vision-language correspondence in image-text interleaved web documents. The reviewers also acknowledge the introduced benchmarks—AnyCIR, SeqCIR, and CSR—for evaluating omni-modality understanding. The model excels in performance on these benchmarks, and an ablation study highlights the importance of each modality in the pipeline.

However, in the initial review, several concerns and questions were raised regarding the practicality and motivation behind unifying text and images into pixel space, especially when they are naturally separate in many cases. There were also common concerns regarding the evaluation tasks on other VL benchmarks beyond retrieval and pure text MTEB. Issues with documentation and presentation were pointed out as well. After rebuttal, the authors addressed many of these concerns. However, regarding the practicality and motivation, one reviewer remains unconvinced.

After carefully reviewing all comments, rebuttals, and discussions, my main concern remains the narrow evaluation task, which was also pointed out by JeXw and MsPN. Besides the evaluation on the text-only task MTEB, the proposed method is inferior to OpenCLIP-T, as reported. Other evaluations are solely on multimodal retrieval tasks, which hardly suffice for VL models. It would be important to include more application scenarios, such as those mentioned by the authors (e.g., Unified-IO, Vlmo). Given these points, I suggest a rejection.

**Additional Comments On Reviewer Discussion:**

The original review mostly comments on the motivation behind unifying text and images into pixels, the lack of other common evaluation benchmarks, questions about the evaluation methods used, and issues with presentation, such as missing details on data augmentation. Additionally, the terms “omni-modality” and “rendering” are confusing. The authors responded by providing further clarification, and while many points were addressed, some concerns remain. For instance, the issue regarding the unclear motivation for unifying text and images into pixels still persists, as noted by reviewer naQ8. However, the most concerning issue to me is the narrow evaluation focusing solely on the retrieval use case, which limits the impact of this work. The authors argue that existing benchmarks, like VQAv2, primarily focus on short sentences, which is not the focus of this paper. While this is true, there is no support provided for the broader application of this work.

---

### Decision · Program_Chairs · 2025-01-22

Reject